# Interpreting the Linear Structure of Vision-language Model Embedding Spaces

**Isabel Papadimitriou**[*][a]    **Huangyuan Su**[*][a,b]    **Thomas Fel**[*][a]
**Sham Kakade**[a,b]    **Stephanie Gil**[b]

[a]Kempner Institute for the Study of Natural and Artificial Intelligence at Harvard University
[b]Department of Computer Science, Harvard University
 {isabelpapadimitriou,csu,tfel,sham,sgil}@g.harvard.edu

## Abstract

Vision-language models encode images and text in a joint space, minimizing the distance between corresponding image and text pairs. How are language and images organized in this joint space, and how do the models encode meaning and modality? To investigate this, we train and release sparse autoencoders (SAEs) on the embedding spaces of four vision-language models (CLIP, SigLIP, SigLIP2, and AIMv2). SAEs approximate model embeddings as sparse linear combinations of learned directions, or "concepts". We find that, compared to other methods of linear feature learning, SAEs are better at reconstructing the real embeddings, while also able to retain the most sparsity. Retraining SAEs with different seeds or different data diet leads to two findings: the rare, specific concepts captured by the SAEs are liable to change drastically, but we also show that commonly-activating concepts are remarkably stable across runs. Interestingly, while most concepts activate primarily for one modality, we find they are not merely encoding modality per se. Many are almost orthogonal to the subspace that defines modality, and the concept directions do not function as good modality classifiers, suggesting that they encode cross-modal semantics. To quantify this bridging behavior, we introduce the *Bridge Score*, a metric that identifies concept pairs which are both co-activated across aligned image-text inputs and geometrically aligned in the shared space. This reveals that even single-modality concepts can collaborate to support cross-modal integration. We release interactive demos of the SAEs for all models, allowing researchers to explore the organization of the concept spaces. Overall, our findings uncover a sparse linear structure within VLM embedding spaces that is shaped by modality, yet stitched together through latent bridges—offering new insight into how multimodal meaning is constructed.

## 1 Introduction

How do vision-language models (VLMs) organize their internal space in order to relate text and image inputs? Multimodal models encode images and text in a joint space, enabling many impressive downstream multimodal applications. In this paper, we explore how multimodal models encode meaning and modality in order to try and understand how cross-modal meaning is expressed in the embedding space of vision-language models.

We approach this question through dictionary learning: the class of methods that consists of finding linear directions (or, "concepts") in the latent space of the model, that can break down each embedding into a linear combination of more interpretable directions. We train multiple dictionary learning methods on top of the four VLMs under consideration (CLIP (Radford et al., 2021), SigLIP (Zhai et al., 2023) SigLIP2 (Tschannen et al., 2025)

---

* These authors contributed equally to this work.

and AIMv2 (El-Nouby et al., 2024; Fini et al., 2024)), and settle on BatchTopK Sparse Autoencoders (SAEs) as the method that dominates the frontier of the tradeoff between remaining faithful to the original embedding space while being maximally sparse. Using the concepts extracted by BatchTopK SAEs, we run a series of investigations to understand how vision-language embedding spaces work. Our analysis is grounded in an ambitious empirical effort: we train and compare SAEs across four distinct vision-language models, using hundreds of thousands of activations, and carry out a thorough suite of evaluations that dissect the geometric, statistical, and modality-related structure of their embedding spaces.

- We show that SAEs trained with different seeds and with different training data mixtures are **stable and robust when we consider the concepts that are commonly used** (high-energy concepts), but very unstable on the concepts that are used very rarely (low-energy concepts) (Section 4).
- We show that **almost all concepts are single-modality concepts**: they activate almost exclusively either for text or for image concepts, and this holds across all models that we try (Section 5.1)
- However, we also show that this does not mean that the concepts lie within the separate cones of the image and text activations: across models, a large proportion of **concepts are almost orthogonal (but not totally orthogonal) to the subspace that encodes modality**, and therefore encode largely cross-modal dimensions of meaning (Section 5.2).
- We develop VLM-Explore (`https://vlm-concept-visualization.com/`), an interactive visualization tool that lets researchers explore the linear concepts in a model, and how they connect modalities (Section 6).

Overall, our findings show how the linear structure of vision-language embedding space can be used to understand the mechanics of joint vision-language space, and we provide the SAEs and the visualization tools for researchers to further explore these connections. Our exploration into the workings of VLM spaces reveals an interesting two-sided conclusion: while the space is organized primarily in terms of modality, single-modality can still be related through high cosine similarities on the subspace that is orthogonal to modality, creating cross-modal bridges of meaning.

## 2    Background and related work

**Linear Representations in Neural Networks**    The representation spaces of neural networks have been often shown to encode many important features (like syntactic or object-category information) largely linearly (Alain & Bengio, 2016; Hewitt & Manning, 2019; Belinkov, 2022; Li et al., 2016; Conneau et al., 2018; Fel et al., 2024a), since, after all, the operations on these spaces are largely linear In vision-language models, strong performance on zero-shot tasks suggests that their embeddings are structured along meaningful directions. Empirical evidence from probing and representation analyses further supports the idea that semantic content is linearly organized Muttenthaler et al. (2023; 2024); Nanda et al. (2021); Tamkin et al. (2023); Bricken et al. (2023).

**Sparse Coding and Dictionary Learning**    Dictionary Learning (Tošić & Frossard, 2011; Rubinstein et al., 2010; Elad, 2010; Mairal et al., 2014; Dumitrescu & Irofti, 2018) has emerged as a foundational framework in signal processing and machine learning for uncovering latent structure in high-dimensional data. It builds upon early theories of Sparse Coding, developed in computational neuroscience Olshausen & Field (1996; 1997); Foldiak & Endres (2008), where sensory inputs were modeled as sparse superpositions of overcomplete basis functions. The core objective is to recover representations where each data point is approximated as a *linear* combination of a small number of dictionary atoms Hurley & Rickard (2009); Eamaz et al. (2022). This sparsity constraint encourages interpretability by associating individual basis vectors with meaningful latent components. In neural network interpretability, sparse decompositions have gained traction as tools to extract interesting features from learned representations (Elhage et al., 2022; Cunningham et al., 2023; Fel et al., 2023; 2024b; Rajamanoharan et al., 2024; Gorton, 2024; Surkov et al., 2024).

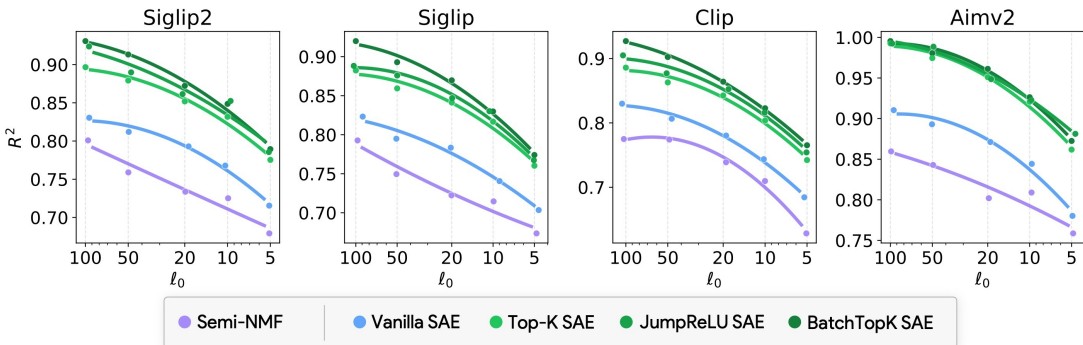

Figure 1: **Selecting a sparse dictionary learning method: the Expressivity-Sparsity trade-off.** Pareto fronts for five dictionary learning methods applied to four vision-language models (CLIP, SigLIP, SigLIP 2, AIMv2). Each curve shows the trade-off between reconstruction quality ($R^2$ score) and sparsity level ($\ell_0$ norm of $\mathbf{Z}$). The three SAEs (TopK, JumpReLU, and BatchTopK) consistently achieve the best balance, with BatchTopK slightly dominating other sparse autoencoder variants.

Sparse autoencoders (SAEs) (Makhzani & Frey, 2014) are a scalable instantiation of Sparse Dictionary Learning implemented by training a small neural network. Recent work has shown that SAEs extract semantically meaningful concepts (Bricken et al., 2023; Bussmann et al., 2024; Thasarathan et al., 2025; Paulo & Belrose, 2025; Gao et al., 2025; Bhalla et al., 2024; Fel et al., 2025), often yielding more interpretable and compositional decompositions than PCA or ICA (Bussmann et al., 2024; Braun et al., 2024; Chen et al., 2024; Fel et al., 2024b; Makelov et al., 2023).

**Modality and Interpretability in Multimodal Models** A growing body of work has examined how multimodal models encode meaning and modality in their internal representations. Despite being trained to align modalities, vision-language models exhibit a persistent *modality gap*, where text and image embeddings occupy distinct conical regions in the joint space (Liang et al., 2022; Shukor & Cord, 2024). However, examining the neurons (Goh et al., 2021; Shaham et al., 2024), representations (Bhalla et al., 2024; Parekh et al., 2024; Wu et al., 2024) shows that multimodal models showcase aspects of cross-modal internal processing.

## 3  Methods: Sparse Autoencoders on Vision-Language Models

### 3.1  Background and Notation.

We denote vectors by lowercase bold letters (e.g., $\mathbf{x}$) and matrices by uppercase bold letters (e.g., $\mathbf{X}$). We write $[n]$ to denote the index set $\{1, \ldots, n\}$. The unit $\ell_2$-ball in $\mathbb{R}^d$ is defined as $\mathcal{B} = \{\mathbf{x} \in \mathbb{R}^d \mid \|\mathbf{x}\|_2 \leq 1\}$. We consider a general multimodal representation learning setting in which a vision-language model maps image or text inputs $\mathbf{x} \in \mathcal{X}$ into a shared representation space $\mathcal{A} \subseteq \mathbb{R}^d$. A collection of $n$ such embeddings is stored in a matrix $\mathbf{A} \in \mathbb{R}^{n \times d}$. Dictionary Learning seeks to approximate $\mathbf{A}$ as a linear combination of concept vectors from a dictionary $\mathbf{D} \in \mathbb{R}^{c \times d}$, using a sparse code matrix $\mathbf{Z} \in \mathbb{R}^{n \times c}$:

$$(\mathbf{Z}^\star, \mathbf{D}^\star) = \arg \min_{\mathbf{Z}, \mathbf{D}} \|\mathbf{A} - \mathbf{Z}\mathbf{D}\|_F^2 \tag{1}$$

where $\|\cdot\|_F$ denotes the Frobenius norm, and dictionary atoms are constrained to lie on $\mathcal{B}$.

In the case of **Sparse Autoencoders (SAEs)**, an encoder network $\boldsymbol{\psi}$, a single-layer MLP, maps embeddings $\mathbf{A}$ to sparse codes $\mathbf{Z}$ via a linear transformation and a projection (Hindupur

et al., 2025) operator $\mathbf{\Pi}\{\cdot\}$:

$$Z = \psi(A) = \mathbf{\Pi}\{AW + b\} \tag{2}$$

where $W \in \mathbb{R}^{d \times c}$ and $b \in \mathbb{R}^c$ are learnable encoder parameters. In particular, we consider several SAEs that can be described by their projection operators such as:

$$\mathbf{\Pi}\{x\} = \begin{cases} \mathrm{ReLU}(x) = \max(\mathbf{0}, x) \\ \mathrm{JumpReLU}(x) = \max(\mathbf{0}, x - \theta) + \theta \odot H(x - \theta) \\ \mathrm{TopK}(x, k) = \arg\min_{z \in \mathbb{R}^c} \|z - x\|_2^2 \quad \text{s.t.} \quad \|z\|_0 = k, \ z \geq 0 \end{cases} \tag{3}$$

Where $H$ is the Heaviside step function. We will also explore a simple extension, *BatchTopK*, which flexibly shares applies the TopK operation across the entire batch.

## 3.2 Selecting a sparse dictionary method: Expressivity vs sparsity

An ideal dictionary learning method should produce representations that are both **expressive** and **sparse**. In practice, this means that model embeddings should activate only a small number of meaningful concepts, indicating that the learned dictionary is both specific and semantically aligned with the data. In our first set of preliminary experiments, we compare five methods on this trade-off, and present our results in Figure 1.

**Dictionary learning methods:** We test five different dictionary learning methods: Semi-Nonnegative Matrix Factorization (Semi-NMF) (Parekh et al., 2025; Lee & Seung, 1999), Sparse Autoencoders (SAEs), Top-K SAEs (Gao et al., 2025), JumpReLU SAEs (Rajamanoharan et al., 2024), and BatchTopK SAEs (Bussmann et al., 2024), and train using Overcomplete.

**Models:** We run experiments on four models: CLIP (Radford et al., 2021), SigLIP (Zhai et al., 2023), and SigLIP2 (Tschannen et al., 2025), which are encoder models, and AIMv2 (El-Nouby et al., 2024; Fini et al., 2024), which is an autoregressive vision-language model

**Data:** We train all dictionary learning methods on reconstructing the activation matrix $A$, consisting of 600,000 normalized embeddings from passing the COCO dataset (Lin et al., 2014) through the models, with each $A_i \in \mathcal{B}$, the unit $\ell_2$-ball

**Results:** We evaluate the expressivity and sparsity of all methods, and present our results in Figure 1. We measure sparsity as the number of non-zero entries in the code vector (i.e., the $\ell_0$ of $Z$). For different levels of sparsity, we plot the reconstruction quality (expressivity): the $R^2$ score between the original activations $A$ and their reconstructions $ZD$. The trade-off captures how well the learned codes retain information while encouraging interpretability through sparsity.

Our results indicate that all four SAE variants outperform Semi-NMF at fixed sparsity levels, with BatchTopK achieving slightly better performance overall. In practice, we also find that training TopK and BatchTopK SAEs is more stable and requires less hyperparameter tuning. For these reasons, **we use the BatchTopK SAEs for our subsequent analysis experiments**.

## 3.3 Metrics: The geometry and statistics of learned concepts

To assess the interpretability and quality of the learned representations, we introduce four complementary metrics that go beyond the standard sparsity–reconstruction trade-off: **energy**, **stability**, **modality score**, and **bridge score**. Each of these measures captures a different axis of representational information:

**Energy.** Energy is a measure of how often each concept is used: its average activation strength across a dataset. Formally, for a concept $i$, we define its energy as: $\mathrm{Energy}_i = \mathbb{E}_z(z_i)$. Energy provides a *statistical* view over the dataset, revealing where attention should be focused for interpreting the learned representation.

---

https://github.com/KempnerInstitute/overcomplete

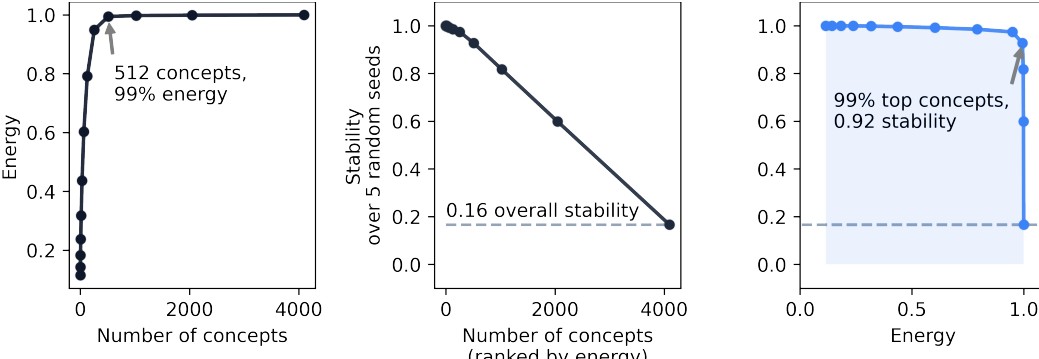

Figure 2: **The concepts that use most of the energy are stable** Here we compare the relationship between energy and stability: (Left) Energy is concentrated on a few concepts, with 512 concepts getting 99% of the energy, and the other 3,500 concepts only appearing in 1% of the total coefficient weight. (Center) When training with 5 different random seeds, the stability when we consider all 4,096 concepts is low – 0.16 (Right) When we weight concepts by energy, we can see that the concepts that are used often in reconstruction are actually stable, with stability of 0.92 if we take the top 512 energetic concepts from each run. **The instability comes from concepts that are rarely if ever actually used**.

**Stability.**   Stability quantifies how consistent the learned dictionaries are between two runs. If two independently trained SAEs on the same model produce significantly different dictionaries, it suggests that the learned concepts are not robust, which weakens their interpretability. Following Fel et al. (2025) and Spielman et al. (2012), we compute the stability between two dictionaries $D, D' \in \mathbb{R}^{c \times d}$ by optimally aligning their rows using the Hungarian algorithm, which solves for a permutation matrix $P \in \mathcal{P}(c)$ that maximizes the total similarity between matched concept vectors:

$$\text{Stability}(D, D') = \max_{P \in \mathcal{P}(c)} \frac{1}{c} \text{Tr}(D^\top P D'). \tag{4}$$

A higher stability score indicates that concept representations are reproducible across training runs, reinforcing their semantic reliability.

**Modality Score.**   In the context of VLMs, a central question is whether a concept is modality-specific or shared across modalities. To probe this, we introduce the modality score, which quantifies how much a concept contributes to reconstructing image versus text inputs. Let $\iota$ and $\tau$ be the empirical distributions of sparse codes $z$ from image ($\iota$) and text ($\tau$) inputs, respectively. We define the modality score of a concept $i$ as the fraction of expected activation energy assigned to image-based inputs:

$$\text{ModalityScore}_i = \frac{\mathbb{E}_{z \sim \iota}(z_i)}{\mathbb{E}_{z \sim \iota}(z_i) + \mathbb{E}_{z \sim \tau}(z_i)}. \tag{5}$$

A modality score near 1 indicates that the concept fires only image inputs, while a score near 0 suggests text dominance. A value around $\frac{1}{2}$ indicates a multi-modal concept.

**Bridge Score.**   While the modality score offers a per-concept perspective, many vision–language tasks rely on coordinated interactions between concepts across modalities. To capture these interactions, we introduce the **bridge matrix** $\mathbf{B} \in \mathbb{R}^{c \times c}$, which quantifies how pairs of concepts contribute to the alignment of modalities. Let $(z_\iota, z_\tau) \sim \gamma$ be a pair of sparse codes obtained from a matching image–text input pair, drawn from a joint empirical distribution $\gamma$ over aligned data points. Let $D \in \mathbb{R}^{c \times d}$ denote the shared dictionary. The bridge matrix is defined as:

$$\mathbf{B} = \underbrace{\mathbb{E}_{(z_\iota, z_\tau) \sim \gamma}(z_\iota^\top z_\tau)}_{\text{co-activation}} \odot \underbrace{(DD^\top)}_{\text{alignment}}, \tag{6}$$

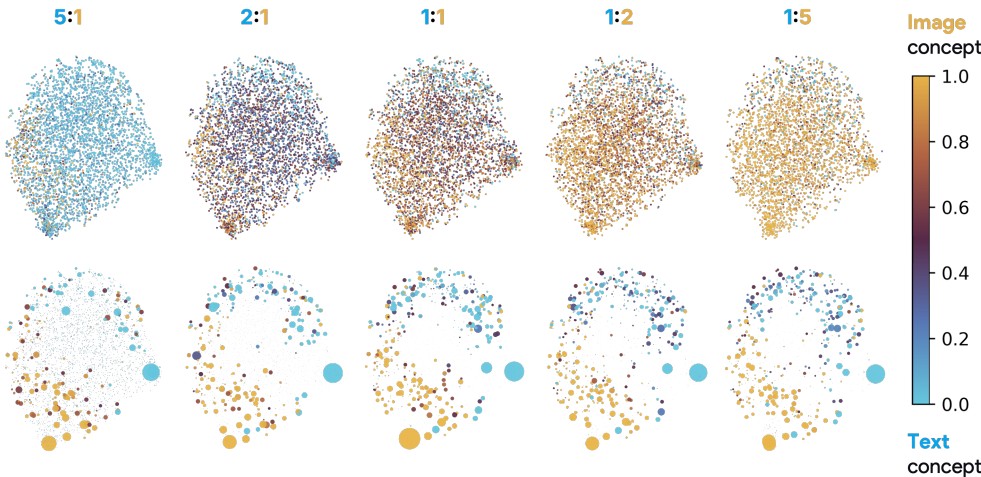

Figure 3: **The geometry of high-energy concepts is stable across data mixtures**. Top: each concept has equal size, Bottom: size is dependent on energy. UMAP visualization of the SAE concept spaces under different image–text data mixtures. Color indicates the modality score of each concept. While the dominant modality in the training data strongly influences how many concepts are recovered per modality (*top row*), the most energetic concepts (*bottom row*) remain relatively stable across mixtures. The set of high-energy text and image concepts remains relatively consistent regardless of the input distribution. Additional results for the other three models are provided in Appendix A, Figure 9.

with $\odot$ being the Hadamard product. The first term captures *statistical* co-activation between concepts: how frequently concept $i$ in the image code and concept $j$ in the text code are simultaneously active for semantically aligned inputs. The second term reflects the *geometric* alignment between the corresponding dictionary atoms, computed as the cosine similarity between dictionary atoms The elementwise product combines both aspects—activation and alignment—into a single interpretable structure.

The resulting matrix **B** reveals how concepts jointly operate across modalities. **In this sense, the bridge matrix provides insight into why the model succeeds at cross-modal alignment**. It highlights not just which concepts are shared, but how they collaborate structurally and statistically to support the model's semantic integration of vision and language.

## 4    Analysis 1: Are We Finding Consistent Model Features?

Before analyzing the geometry and semantic content of the concepts extracted by our SAEs (Section 5), we first examine whether these concepts are **consistent** and stable: do we get similar concepts across different conditions? We study stability across: *(i)* random seeds—to test sensitivity to training stochasticity, and *(ii)* data modality mixtures—to test dependence on the distribution of input activations.

### 4.1    The concepts that use most of the energy are stable

We train five SAEs with different random seeds on the same SigLIP2 activations, each using 4096 concepts, and compute the average pairwise stability (Equation 4) between them. As shown in Figure 2 (center), at first glance, overall stability is low: the mean similarity between dictionaries is just 0.16, suggesting high sensitivity to random initialization. However, this apparent instability is misleading. On the left, see that reconstruction energy is heavily concentrated in a small subset of concepts: the top 512 concepts account for approximately 99% of the total activation mass. To isolate the functionally relevant features, we recompute stability using only the top-$k$ most energetic concepts from each run. We find that **high-energy concepts are remarkably stable**, with near-perfect alignment across

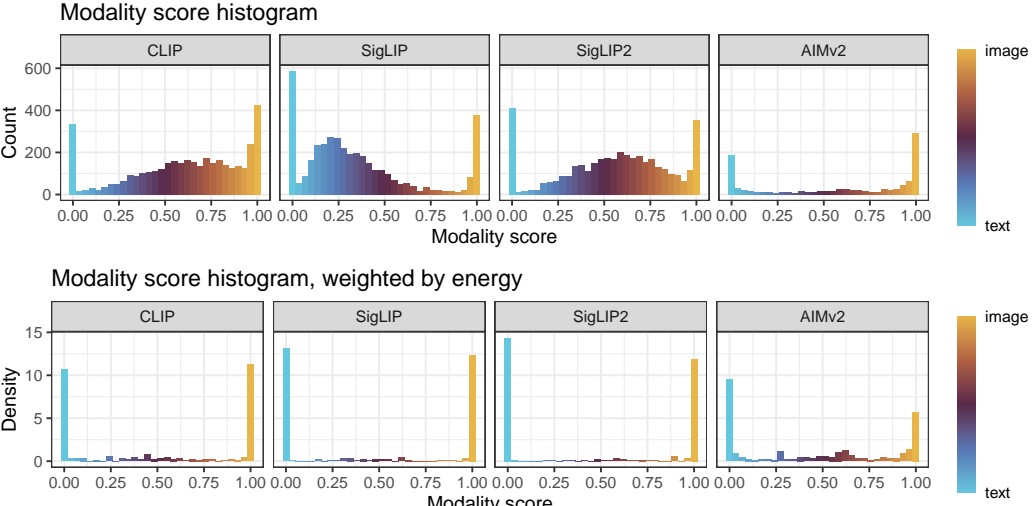

Figure 4: **Most concepts are single-modality** Histograms of the modality scores (top) and the modality scores weighted by energy (bottom) of every concept in the SAEs. On every model, the modes are the two extremes: concepts either activated only by text or only by image. Weighting by energy makes this much more prominent, showcasing that almost all of the reconstruction coefficients go to concepts that are single-modality.

seeds (Figure 2, right). The instability arises almost entirely from the low-energy tail of the dictionary—concepts that contribute little to the reconstruction objective. We conclude that SAEs trained with different seeds reliably recover a core set of functional, high-usage concepts.

## 4.2 The geometry of high-energy concepts is stable across data mixtures

Though we have seen that the high-use concepts retrieved by the SAE are stable across random initializations, there is still the question of how much the recovered concepts are influenced by the data used to train the SAEs. Do our SAE dictionaries reflect the structure in the model or simply the statistics of the dataset used to train the SAE? To test this, we train SAEs varying the **modality mixture** of the training data: how many of the activations we use to train the SAE come from image inputs vs. text inputs. We train SAEs on seven data compositions, ranging from heavily image-dominant (5:1 image:text) to heavily text-dominant (1:5). Results are shown in Figure 3, where we contrast the UMAP geometry of the learned concepts concepts (top) with the geometry of the concepts when the size of each point is weighted by the concept's energy. As expected, changing the data mixture shifts the overall modality distribution of the learned concepts: the text-heavy SAEs learn more text concepts (blue points) and the image-heavy SAEs learn more image concepts. However, when weighting by energy (bottom row), the **geometry of the most energetic concepts remain stable across all mixtures**: the visualizations look remarkably consistent across data mixtures. High-energy directions consistently appear regardless of the training distribution, and exhibit similar activation patterns.

**Discussion: Why low-energy concepts?** Our evaluations show that different SAEs discover widely different low-energy features. Does Does this mean that low-energy concepts are useless? Not necessarily. Leveraging VLM-Explore (Section 6) we observe that many low-energy concepts correspond to coherent but rare patterns, such as "yaks" or "valentines day hearts." These concepts may be semantically meaningful, but their contribution to reconstruction is small, and thus SAEs do not consistently recover them. By contrast, high-energy concepts typically capture more concepts like "red" or "two women" and are consistently recovered because they support a wide range of reconstructions. We provide some illustrative examples of concepts in Appendix B.

Figure 5: **Image and text activations lie in separate cones**. We plot the distribution of cosine similarities between model activations. Image–image and text–text pairs (blue/green) are close, while image-text pairs (orange) are much farther, indicating that embeddings occupy separate modality-specific cones.

These analyses show that while SAE dictionaries may appear unstable, this instability is concentrated in low-energy concepts. The high-energy subset—responsible for nearly all reconstruction—is both robust to random initialization and is also not simply a reflection of the data. These findings guide the rest of this paper. They suggest that SAEs can be used as reliable tools for interpreting VLM embeddings—**but only when conditioned on energy**. Concepts with high energy form a stable and informative basis; concepts with low energy may still be meaningful, but should be treated with caution due to their instability.

## 5 Analysis 2: Are We Finding Features With Cross-Modal Meaning?

Having established that SAEs recover stable and consistent high-energy features, we now ask whether these features capture structure that spans both modalities. Specifically, we analyze whether the learned concepts reveal aspects of the geometry of the joint vision-language embedding space, and whether they support cross-modal alignment.

### 5.1 Most concepts are single-modality

Our first observation is that the majority of concepts activate predominantly on a single modality—either text or image. As defined in Equation 5, the modality score of a concept measures the fraction of its total activation energy that comes from image inputs. A score near 1 indicates image-specific usage; a score near 0 indicates text-specific usage. In Figure 4, we plot the modality score distribution across all concepts for SAEs trained on each model. The top row shows raw counts: while some concepts cluster around 0.5, the distributions are clearly bimodal, with dominant modes at 0 and 1. This pattern becomes more pronounced when we weight each concept by its energy (bottom): almost all of the energy across all models is concentrated on concepts that are nearly exclusive to one modality.

We conclude **that the most influential directions in the SAE dictionary — those responsible for the bulk of reconstruction — are overwhelmingly single-modality**, despite the shared embedding space.

### 5.2 Image and text activations lie in separate cones — but many concepts are orthogonal to this difference

In all models that we analyze, we find that the activations of image and text inputs lie in separate, narrow cones. To measure this, we compare the cosine distance between activations of different modalities to the distance between activations of the same modality. If the average cosine similarity between a set of points is significantly different from zero, this means that the points lie in a narrow cone (with respect to the origin), rather than being distributed around the unit ball evenly. Following findings by Tyshchuk et al. (2023) (as well as Mimno & Thompson (2017) and Ethayarajh (2019) in the language-only context), we find that **image and text activations lie on separate, narrow cones**. We present our results in Figure 5: embeddings of the same modality (green, blue) are significantly closer than those of different modalities (orange), which are on average almost orthogonal. This suggests that image and text activations reside in distinct conical regions of the embedding space.

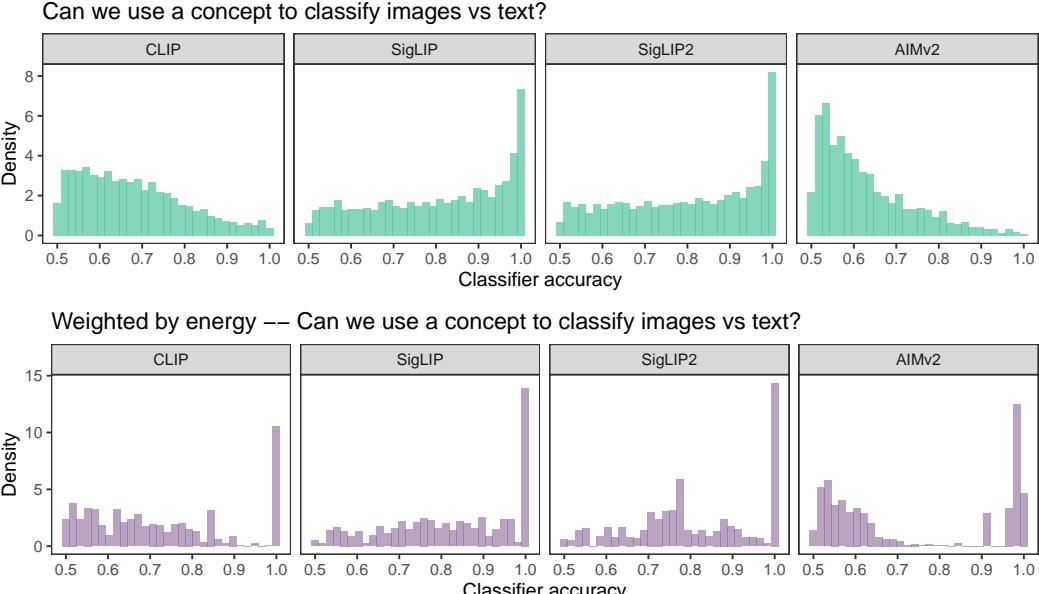

Figure 6: **Many concepts are not aligned with the modality directions**. Histogram of the accuracies of concepts used as classifiers (on the bottom each concept is weighted by its energy). An accuracy of 0.5 means that the concept is orthogonal to every linear direction that defines modality (and therefore likely encodes some aspect of meaning independent from modality), and an accuracy of 1 means that the concept is aligned with a modality direction. We see that, especially in CLIP and AIMv2, there is a significant proportion of concepts that are almost orthogonal to the modality subspace. When we weight by energy, we see that accuracy-1 concepts are high-energy, but that concepts less aligned with modality still receive a significant amount of energy.

Do the concept vectors align with the modality structure? If they follow the geometry of the modality cones that separate image from text, they likely encode little cross-modal content. But if many lie near directions *orthogonal* to the modality subspace, they may instead reflect shared meaning. We test this by testing **how well each concept performs as a linear modality classifier**. To do this, we project input activations onto the direction defined by the concept, and measure how well it separates image from text. High accuracy implies alignment with modality; near-chance performance suggests the concept is modality-agnostic and may encode cross-modal structure. Figure 6 presents the distribution of classification accuracies for all concepts (left) and weighted by energy (right). While many concepts—particularly high-energy ones – are highly predictive of modality, a substantial number achieve near-random accuracy. These findings suggest that many concepts *geometrically close* to a modality-agnostic subspace, even though they activate for a specific modality This points to the existence of sparse linear directions that support semantic alignment across modalities while remaining sensitive to modality itself.

### 5.3 Proposal: Cross-modal alignment in VLM linear concepts

Our results lead us to a proposal of how the linear directions found by SAEs in multimodal space function, of which we provide a schematic in Figure 7

1. The image and text activations in the VLMs we study lie in separate, narrow cones (Figure 5), and SAE concepts tend to **activate primarily for one modality** (Figure 4).

2. However, SAE concepts largely **do not lie along the linear subspace** that separates the image and text cones (Figure 6), and in practice they can express semantics *bridge* **between modalities**. The Bridge Score (Equation 6) shows how there are

many-to-many relationships between related concepts in text and image space (see Appendix B, Figure 16 for an illustrative example)

3. How can we resolve this tension? The resolution lies in considering the **SAE projection effect** (Figure 8): due to the sparsity constraints (like TopK), SAE concepts only activate for a tiny number of the input activations that they are well-aligned with. If an SAE concept is almost orthogonal to the modality-separating subspace, but leans slightly towards text, it is very unlikely to ever be in the top-K concepts for an image input. Though the concept will seem like a text-only concept, it is in fact aligned with many image outputs.

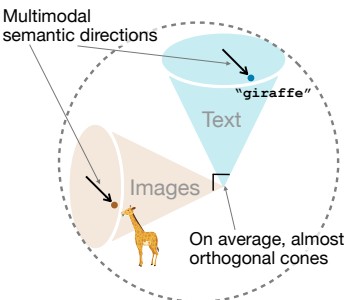

Figure 7: A schematic of our proposal in a toy 3D space. Though images and text can lie in separate orthogonal cones with respect to some subspace (in this case the x-y plane), cross-modal directions can emerge (in this case, the z direction coming towards us represents the cross-modal semantics of "giraffe"). In large latent spaces, many dimensions can form a complex multimodal space that the z-axis is standing in for here.

Figure 8: The SAE projection effect: We plot the dot product of a set of text (blue) and image (orange) activations with an SAE concept. Even though the concept direction does not separate modality, the TopK thresholding step (dotted line) may consistently select only one modality

## 6 VLM-Explore: An interactive, functional concept explorer

Lastly, we present VLM-Explore (https://vlm-concept-visualization.com), an interactive visualization tool designed to facilitate the exploration and analysis of our SAE concept representations. VLM-Explore offers an intuitive visualization that combines into one interactive figure four important aspects that help us interpret and debug VLMs. These are: 1) the UMAP structure shows how the linear directions of the latent space relate to each other 2) the maximally activating examples panel shows what each linear feature actually represents 3) Each feature is colored according to its Modality Score (Equation 5) showing how the two modalities share the space, and 4) the Bridge Score connections show what what semantics connect the two modalities.

## 7 Discussion and Conclusion

We study the linear structure of vision-language embedding spaces using SAEs trained on four VLMs. While concept dictionaries vary across seeds and data mixtures, the high-energy subset is consistently recovered and accounts for nearly all reconstruction. Most concepts are single-modality in usage but often lie near the modality-agnostic subspace, suggesting shared structure and a modality score shaped by SAE thresholding rather than direction. To analyze cross-modal alignment, we introduce the Bridge Score, identifying concept pairs that are both geometrically aligned and co-activated. We release VLM-Explore, an interactive tool to visualize these structures across models.

These insights lay a foundation for building more interpretable and controllable multimodal models, and open the door to future research into structured alignment mechanisms.

## Acknowledgments

This work has been made possible in part by a gift from the Chan Zuckerberg Initiative Foundation to establish the Kempner Institute for the Study of Natural and Artificial Intelligence at Harvard University.

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

## A  Additional Figures and Expanded Discussion

### A.1  UMAP Visualizations Across Data Mixtures for all models

Figure 9 shows UMAP projections of the learned SAE concept dictionaries across different training data mixtures. Each subfigure corresponds to a different VLM (SigLIP2, SigLIP, CLIP), with UMAP applied to the learned dictionary atoms. These visualizations complement the main Figure 3, now extended to multiple models.

The top rows present concepts with uniform dot size, providing an unweighted view of the dictionary's structure. Here, modality is encoded by color reflecting the modality score (as defined in Eq. 5 of the main paper). The bottom rows rescale each concept's dot size according to its energy score, revealing the relative contribution of each concept to the reconstruction task.

Our key observations include:

- Across all models, when data mixtures are skewed (e.g., more image than text inputs), the distribution of concepts by modality shifts accordingly. This confirms the sensitivity of low-energy, rarely-used concepts to training data statistics.
- However, the bottom rows show that energy-dominant (i.e., frequently activated) concepts remain consistently placed in similar areas of the space, regardless of training mixture. These concepts represent a stable semantic core that is robust to dataset composition.

This supports our conclusion from Section 4 that the SAE models recover a stable, high-energy subspace, even under strong shifts in input distributions.

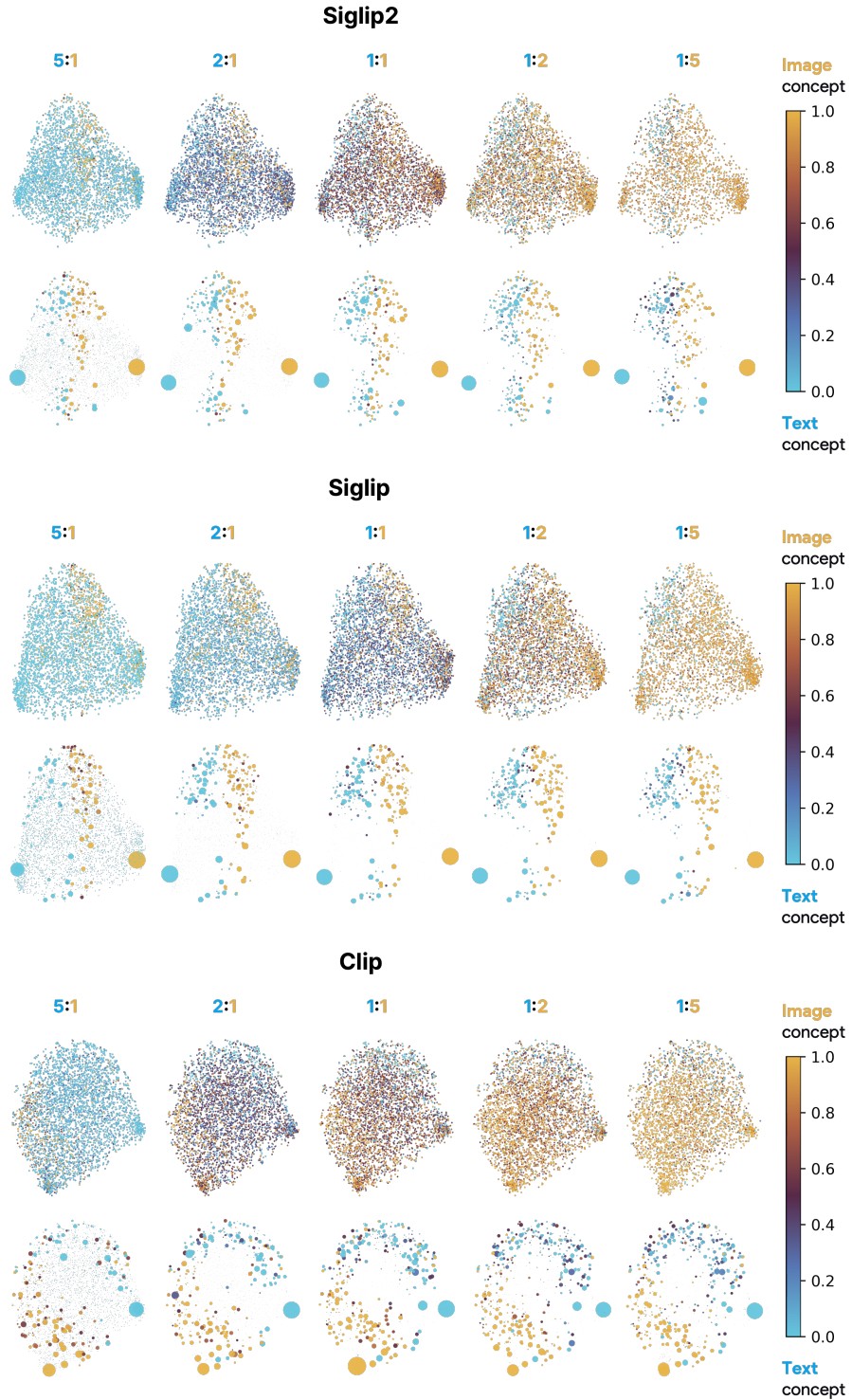

Figure 9: **UMAP projections** of SAE concept dictionaries across various image-text training mixtures for **SigLIP2**, **SigLIP**, and **CLIP**. Top row: equal dot size. Bottom row: dot size scaled by energy. Related to Figure 3.

# B  Illustrative Examples from the Interactive Demo

## B.1  Examples of concepts

We present some qualitative examples of what kinds of concepts the SAEs that we train in the paper fire for, showing concepts from all 4 models. We showcase concepts of different abstractions, starting from more surface-level concepts and going to more abstract concepts. These concepts can all be accessed by putting their number into the `Specific ID` field in https://vlm-concept-visualization.com/ after selecting the correct VLM from the drop-down.

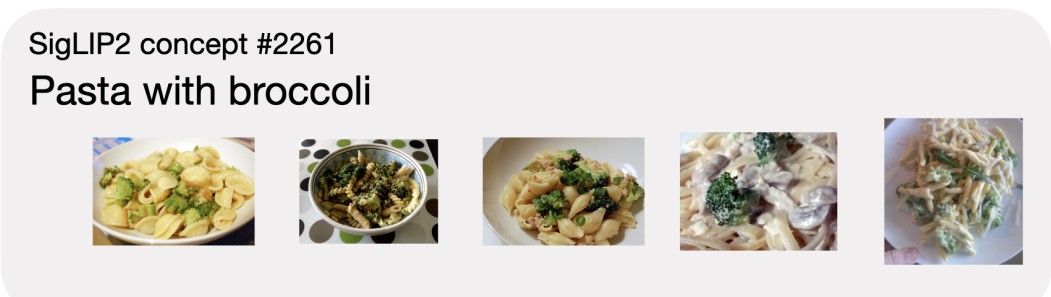

Figure 10: **Coherent concept, with similar visual profile**. Concept #2261 in SigLIP2 fires for pictures of pasta and broccoli, which is an interpretable concept, where all of the images look pretty similar in terms of color, texture, and shape composition

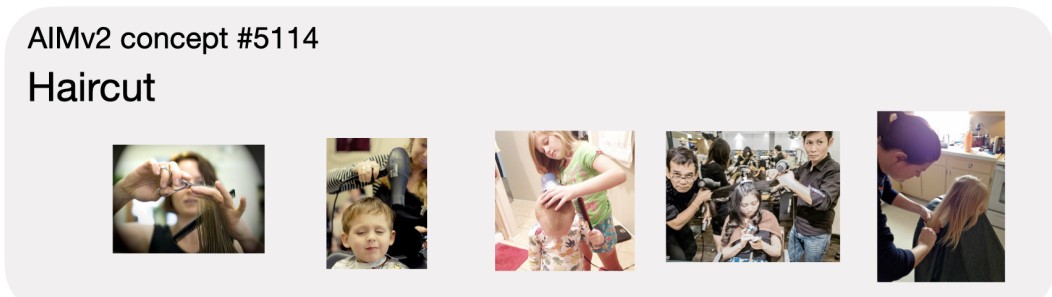

Figure 11: **Coherent concept, with different visual profiles**. Concept #5114 in AIMv2 fires for pictures of people getting haircuts. Though these pictures have fewer surface-level elements that are consistent between them, they all depict the same higher-level interpretable event.

CLIP concept #1033
## Old film photos with rounded edges

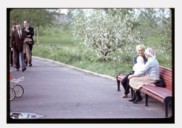 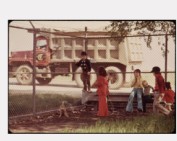 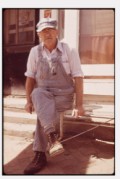 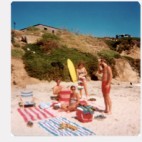 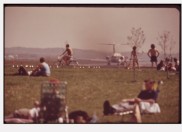

Figure 12: **Similar visual profiles, different semantics**. Concept #1033 in CLIP fires for pictures that have vintage film coloring, rounded edges, and are outdoors. Though the surface-level visual profiles of the top activating examples for this concepts are related, there aren't discernible higher-level semantics that connect them.

SigLIP concept #5812
## Many things

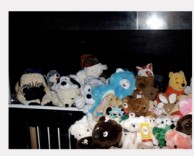 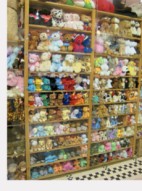 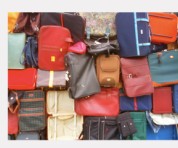 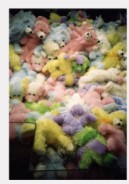 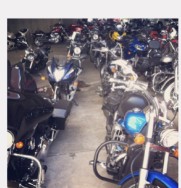

### Highest BridgeScore concepts:

**#4080:** **Large group** of white and black vases sitting next to each other.

**Several** grinder sandwiches with lettuce and tomato and cheese.

**#3231:** **A line of** brief cases sit next to each other

**A line of** elephants that were walking in a line

Figure 13: **Higher-level abstract semantic concept**. Concept #5812 in SigLIP fires for pictures of many disorganized things. The concepts which are most related to it by BridgeScore are text concepts detailing the idea of many things, rather than any other more surface-level semantics of the images (like, "toys"), hinting that the model has learned a **cross-modal representation of the abstract notion of "several"**

**SigLIP concept #2826**

## Blue

A woman wearing a blue outfit
and a blue veil.

A man and women wearing some
odd blue things.

A blue chrome motorcycle with
a dark blue seat.

A blue park bench painted
blue sits on a blue sidewalk.

Two blue chairs sitting next
to a blue wooden table.

Figure 14: **Text concept that is surface-level and visually-directed**. Concept #2826 in SigLIP fires for text that describes blue objects. This is a surface-level text feature, describable by the identity of one token. It is also a feature of the text that relates directly to the visual modality.

**SigLIP concept #1926**

## Hypotheticals, opinions, modals

Maybe the man picked that dog
because they have the same
color hair.

She's probably not going to
get to that frisbee in time.

Perhaps they shouldn't be
playing outside on such a
smoggy day.

She didn't expect that there
would be this many birds to
feed.

Only an idiot would put
mustard on a pair of sugar
frosted donuts.

Figure 15: **Abstract linguistic text concept**. Concept #1926 in SigLIP fires for text that contains uncertainty and/or modal verbs like "would" or "should". This concept is picking up on abstract linguistic features about hypothetical. There is no clear relationship in the visual modality between these captions (unlike the "blue" text), and instead there is a language feature that connects them.

## B.2 Illustration of the BridgeScore

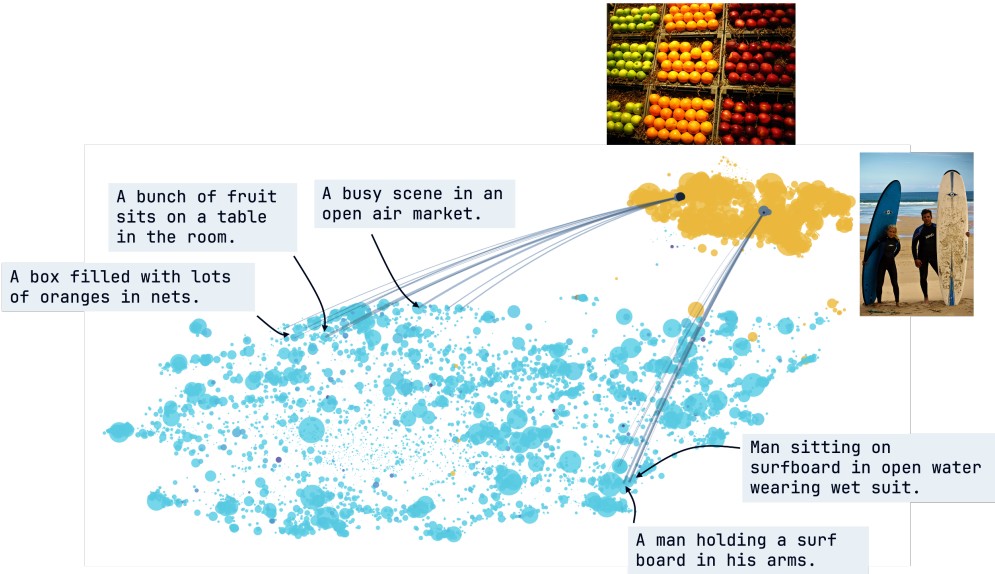

Figure 16: **Bridge score identifies semantically aligned concept pairs across modalities.** Even though most concepts are unimodal in activation, many form meaningful connections across modalities via the bridge score. Edges connect concept pairs that are both geometrically aligned and co-activated on paired examples. These links often reflect many-to-many relationships between clusters of related concepts, enabling cross-modal alignment despite sparse unimodal structure.

## C Training Details for SAEs

We report the full training configuration used for the sparse autoencoders (SAEs) in this work. All SAEs were trained on frozen activations from the COCO dataset, consisting of 600,000 image and text embeddings per model. Each SAE was trained for 30 epochs, with a batch size of 1024, totaling approximately 18 million training examples. The encoder consists of a single linear projection without hidden layers, mapping input activations to a 4096-dimensional concept space. Unless specified, we used sparsity $k = 5$, enforcing exactly five non-zero, non-negative coefficients per input.

Optimization was performed using the AdamW optimizer with a cosine learning rate schedule. The learning rate warmed up from $1 \times 10^{-6}$ to a peak of $5 \times 10^{-4}$, then decayed back to $1 \times 10^{-6}$. We used weight decay of $1 \times 10^{-5}$ and applied gradient clipping with a maximum global norm of 1.0. All model embeddings were $\ell_2$-normalized prior to encoding. For image inputs, the original images were resized to 256 pixels on the shorter side and center-cropped to $224 \times 224$ before encoding by the VLMs. Text inputs were processed using the default tokenizer for each model.

We observed consistent convergence of the reconstruction loss around epoch 10, but trained for 30 epochs to ensure full stability of the learned dictionaries. We used the Overcomplete toolbox for training.

---

https://github.com/KempnerInstitute/overcomplete

