# OpenReview forum: "Interpreting the linear structure of vision-language model embedding spaces"
_colmweb.org/COLM/2025/Conference — COLM 2025_

### Official Review · Reviewer_Hk33 · 2025-05-12

**Rating:** 7
**Confidence:** 3
**Ethics Flag:** 1

**Summary:**

The paper presents the training and release of sparse autoencoders (SAEs) applied to the encoding spaces of four vision-language models: CLIP, SigLIP, SigLIP2, and AIMv2. It demonstrates how the linear structure within the vision-language embedding space can be leveraged to better understand the joint vision-language representation. Additionally, the authors provide the trained SAEs, visualization tools, and introduce new evaluation metrics including stability, modality score, and bridge score to assess the models.

**Questions To Authors:**

1. The rationale behind selecting only encoder-based vision-language models for interpretation is not clearly explained.

2. While the paper states that training TopK and BatchTopK sparse autoencoders (SAEs) offers greater stability and requires less hyperparameter tuning, detailed training procedures are not provided.

3. Providing qualitative analysis to illustrate what constitutes image and text concepts would enhance understanding.


Minor Comments

1. Equation-3 needs to define the parameters used.
2. https://vlm-concept-visualization.com/ only shows Energy as check-box, do not provide any other options? Is it supposed to have only this option?

**Reasons To Accept:**

1. Dictionary learning is employed as a method to identify linear directions, or "concepts," within the model's latent space, enabling each embedding to be represented as a linear combination of more interpretable components.

2. Sparse autoencoders (SAEs) are trained and compared across four different vision-language models using hundreds of thousands of activations, followed by comprehensive evaluations that analyze the geometric, statistical, and modality-related structures of their embedding spaces.

3. The study introduces new metrics-stability, energy, modality, and bridge scores-that extend beyond the traditional sparsity–reconstruction trade-off to better assess the quality and characteristics of the learned representations.

**Reasons To Reject:**

1. Although the paper centers on analyzing embedding spaces of vision-language models, it does not clearly specify which datasets were used to generate the quantitative examples presented in Figure 12.

2. The term "sparse auto-encoders (SAE)" may be misleading, as the method does not create autoencoders in the traditional sense but rather represents embeddings as sparse, weighted sums of linear “concepts.”

3. The study focuses solely on encoder-based vision-language models such as CLIP and SigLIP2, overlooking other notable models like LLaVA and its variants.

---

> ### Author Response · Authors · 2025-06-02
>
> Thank you for your positive review and constructive (and actionable) feedback! We have addressed some of the weaknesses/reasons to reject (W) and your question (Q) below:
>
> **W1: Dataset specification for Figure 12:** You're absolutely right—we used the COCO dataset for all quantitative analyses including Figure 12. We'll clarify this in the revision and ensure all figures clearly specify their data sources.
>
> **W2: SAE terminology**: Fair point about terminology. While we use "autoencoder" following standard SAE literature, you're correct that these are more accurately described as sparse dictionary learning methods. We'll add a clarifying note about this distinction in our methods section.
>
> **W3 & Q1: Encoder-based model focus:** We focused on encoder-based models (CLIP, SigLIP variants) plus one autoregressive model (AIMv2) because they produce explicit joint embeddings optimized for cross-modal similarity. Decoder-based models like LLaVA process modalities differently and would require adapted analysis methods. However, you're right that we should better justify this scope and acknowledge it as a limitation for future work. We did not see the scope of this paper as providing a thorough experimental account  of how architecture and learning objectives influence multimodal representation, but instead to introduce methods, metrics, resources, and interesting outlooks in order to advance interpretability in VLMs and better understand how the cross-modality correspondence can work geometrically. Hopefully, using the methods introduced in this paper, follow-up work on architecture can yield important insights about how to better get deep multimodal representation, which would be extremely interesting!
>
>
> **Q2: Training procedures:** Absolutely, as asked by other reviewers too, we’ve added a dedicated section in the appendix detailing hyperparameter, convergence, training procedure to ensure full reproducibility.
>
> **Q3: Qualitative analysis:** Great suggestion! We're adding concrete examples of discovered concepts with visualizations in our supplementary material to make the abstract "concept" notion more tangible. We’ve included some examples here
>
>  https://vlm-concept-visualization.com/demo-multimodal-concepts/COLM_2025_qualitative_examples.pdf
>
> showing concepts that vary in abstractness from surface-level visual features to more abstract meaning and linguistic features.

---

> ### Comment · Area_Chair_Trrv · 2025-06-07
> **Discussion**
>
> Dear Reviewer Hk33,
>
> The authors have responded to your review. Does their answer address your questions and reasons to reject?

---

### Official Review · Reviewer_VzYu · 2025-05-13

**Rating:** 7
**Confidence:** 3
**Ethics Flag:** 1

**Summary:**

The authors explore the linear structure of the embedding spaces of vision-language models (VLMs) using sparse auto-encoders (SAEs) trained on CLIP, AIMv2. While the specific rare concepts found by these models can change, the frequently used concepts, which are more important, are stable across different training conditions. Even though the vision-language embedding space is of joint modalities, the concepts usually activate for a single modality only. Many concepts are also orthogonal and have cross-modal meaning due to the sparsity projection. They also introduces a score (Bridge Score) to identify concept pairs which are crucial for cross-modal alignment and an interactive visualization tool for it called VLM-Explore.

**Questions To Authors:**

How would the findings of the paper generalize to other VLM architectures (ie not contrastive) or to embeddings from different layers within the models?

**Reasons To Accept:**

- The SAE-based approach for interpreting embedding spaces provides meaningful and interesting insights into concept organization and modality bias.
- It provides robust empirical findings across multiple VLMs w.r.t. stability of high-energy concepts and geometric properties of modality despite unimodal activation.
- Bridge Score and VLM-Explore are useful resources for future research in this space.

**Reasons To Reject:**

- The reliance on SAEs for interpreting VLM structure could be potentially limiting and providing an incomplete picture since it is missing non-linear dynamics.
- The interpretation of the VLM space is inherently limed by the potential limitations of SAEs rather than solely reflecting the VLM's intrinsic structure.

---

> ### Author Response · Authors · 2025-06-02
>
> Thanks for your positive and constructive review! We have addressed some of the weaknesses/reasons to reject (W) and your question (Q) below:
>
> **W1 Missing non-linear dynamics:** Spot on! We believe this to be a really important question the authors debated extensively during the writing process! We justify our linear analysis for three reasons: (i) VLM training objectives explicitly enforce linear separability between aligned/misaligned pairs, (ii) downstream tasks predominantly use linear probes, making linear structure directly relevant to understand what downstream task can capture and (iii) the Linear Representation Hypothesis posits that neural networks can encode up to exp(n) nearly orthogonal feature in space of dimension n, extractable linearly.
>
> Our high reconstruction scores (Figure 1) suggest the LRH is at least partially true here: we can reconstruct the large majority of variance using relatively few linear concepts. While SAEs certainly dictate what we can and cannot see, they capture as much as any linear probe method. Importantly, understanding what linear probes can capture is itself valuable, since this determines what downstream applications can access.
>
> However, you're absolutely right that non-linear features exist and that SAEs can’t extract them (but neither do linear probes). We deliberately title our work "linear structure" to be clear about this scope. This tension between linear vs. non-linear feature extraction in multimodal spaces is something we plan to continue exploring in future work.
>
> **W2 Limited to seeing artifacts of the SAE methodology** This is an excellent and fundamental question. You're right that any interpretability method introduces its own lens, this is true for PCA, NMF, SAEs, or any non-trivial analysis. However, our findings reveal something meaningful about VLM structure regardless of methodology: BatchTopK SAEs achieve the best reconstruction (Figure 1) precisely because VLM spaces have a specific geometric organization. The fact that the optimal reconstruction strategy uses concepts that are almost modality-orthogonal concepts selected in a modality aware way tells us something important about how these spaces are structured.
>
> Our key insight is that VLM spaces exhibit a paradox: they're geometrically organized to separate modalities (shown in our anisotropy experiments, Figure 5), yet the most effective way to reconstruct activations is through a nearly modality agnostic basis. This suggests that semantic meaning in VLMs is encoded orthogonally to modality separation, a finding that emerges from the reconstruction objective, not the SAE methodology per se.
>
>
> **Q1 Generalization to other models and layers** Our analysis spans contrastive (CLIP, SigLIP, SigLIP2) and autoregressive (AIMv2) models, showing consistent patterns despite different training objectives. For other architectures, we expect similar linear structure given the linear representation hypothesis, though specific organization might differ.
>
> Regarding layers: we focused on final embeddings where the joint multimodal space is established. Intermediate layers in most VLMs are unimodal (separate text/image processing), so they wouldn't exhibit the cross-modal structure we analyze. However, studying how unimodal representations transition to multimodal ones could reveal how this joint organization emerges. Furthermore, expanding these analyses to single-backbone architectures could let us see how multimodality emerges throughout layers in those models, and understand if it arises with similar dynamics to other complex processing like shape perception or syntax processing. We think that these are very interesting avenues to future work!
>
> In case it is interesting, we have also added some extra qualitative analyses of our features here:  https://vlm-concept-visualization.com/demo-multimodal-concepts/COLM_2025_qualitative_examples.pdf
>
> Thanks!

---

> ### Comment · Reviewer_VzYu · 2025-06-05
>
> Thank you for the clarification, the rating is appropriate.

---

> > ### Author Response · Authors · 2025-06-07
> > **Thank you**
> >
> > Thank you for the thoughtful review and for taking the time to engage with our work! Your feedback really helped us clarify some important points in the paper. Thanks again!
> >
> > Best,
> > The Authors

---

### Official Review · Reviewer_WVqz · 2025-05-13

**Rating:** 7
**Confidence:** 3
**Ethics Flag:** 2

**Summary:**

This paper investigates the linear structure of the joint embedding space in four Vision-Language Models (CLIP, SigLIP, SigLIP2, AIMv2) using Sparse Autoencoders (SAEs). The authors train SAEs to decompose embeddings into sparse linear "concepts," finding SAEs offer a good sparsity-reconstruction trade-off. Key findings include: high-energy concepts are stable across training runs, while low-energy ones are not ; most concepts are highly modality-specific (activating primarily for text or image); however, many concepts, including energetic ones, are geometrically near-orthogonal to the modality-distinguishing subspace, suggesting they encode cross-modal meaning despite unimodal activation patterns. The authors introduce metrics like Modality Score and Bridge Score, and release an interactive visualization tool (VLM-Explore).

**Questions To Authors:**

- Hyperparameter Tuning: What specific hyperparameters were tuned for the Sparse Autoencoders (SAEs), and what procedure was employed?
- Training Convergence: How was training convergence of the SAEs determined (e.g., specific thresholds or observation of plateaus)?
- Equation 4 (Stability): How does the number of concepts (c) impact the Stability score in Equation 4?
- How might the observed linear structures (concept modality, orthogonality) change when analyzing embeddings derived from more diverse datasets or task-specific fine-tuned VLMs?

**Reasons To Accept:**

- This study applies and compares multiple dictionary learning methods (SAEs vs. Semi-NMF) across four different VLMs.
- Some interesting findings are presented, which reveal nuanced structure, particularly the separation of concepts' modal-specific activation patterns and their near-orthogonal geometric alignment relative to modality.
- It provides trained SAEs and an interactive visualization tool (VLM-Explore) for community use.

**Reasons To Reject:**

- A large family of representation models are missing from the study: A notable omission in this study is the family of VLM-based embedding models that rely on a shared Transformer backbone and pooling module for generating both textual and visual representations. These models are capable of producing high-quality embeddings and might exhibit a different representational structure compared to the examined CLIP-style and autoregressive VLMs. We strongly recommend including these architectures in future revisions, as doing so would broaden the study's coverage of mainstream VLM representations and potentially yield complementary findings.
- The "SAE Projection Effect": The explanation for how concepts can be near-orthogonal yet unimodal relies heavily on the non-linear projection step of the SAE. This raises questions about whether the findings reflect the intrinsic linear structure of the VLM space or an artifact introduced by the SAE methodology itself.
- Scalability/Generalization of Findings: Analysis is based on embeddings from one dataset (COCO). It's unclear if the observed structure (e.g., modality scores, orthogonality) holds across different data distributions or more complex VLM tasks.

[1] Jiang, Ting, et al. "E5-v: Universal embeddings with multimodal large language models." arXiv preprint arXiv:2407.12580 (2024).
[2] Jiang, Ziyan, et al. "Vlm2vec: Training vision-language models for massive multimodal embedding tasks." arXiv preprint arXiv:2410.05160 (2024).
[3] Lin, Sheng-Chieh, et al. "Mm-embed: Universal multimodal retrieval with multimodal llms." arXiv preprint arXiv:2411.02571 (2024).

---

> ### Author Response · Authors · 2025-06-02
>
> **W1 Using shared-backbone VLMs.** Thanks for this suggestion, we think that testing these methods and results on a diverse set of models is definitely a very necessary direction for future work. For this paper, we see our main contribution as being that we introduce methods, metrics, resources, and interesting outlooks in order to advance interpretability in VLMs and better understand how the cross-modality correspondence can work geometrically, rather than give a thorough experimental account of how architecture and learning objectives influence multimodal representation
>
> We run all of our analyses on four models: three models trained with contrastive objectives (CLIP, SigLIP, SigLIP2), and one autoregressive model (AIMv2). This lets us demonstrate the kind of generally consistent insights that we can gain when we use our methods, but of course such work cannot claim to be uncovering some truth that holds for all VLMs. As you say, for that, we would  need to test much more exhaustively
>
> You are definitely right, though that we should talk about the whole space of VLMs, and we will take care and set out and describe the diversity of VLMs better in the introduction and background, in order to make it clear that we are only testing on a specific subset of four. We’ll be sure to do this in revisions, and we thank you very much for the citation pointers!
>
> **W2 Are we just seeing artifacts of the SAE methodology?** This is an excellent and fundamental question. You're right that any interpretability method introduces its own lens, this is true for PCA, NMF, SAEs, or any non-trivial analysis.
>
> However, our findings reveal something meaningful about VLM structure regardless of methodology: BatchTopK SAEs achieve the best reconstruction (Figure 1) precisely because VLM spaces have a specific geometric organization. The fact that the optimal reconstruction strategy uses concepts that are almost modality-orthogonal concepts selected in a modality aware way tells us something important about how these spaces are structured.
>
> Our key insight is that VLM spaces exhibit a paradox: they're geometrically organized to separate modalities (shown in our anisotropy experiments, Figure 5), yet the most effective way to reconstruct activations is through a nearly modality agnostic basis. This suggests that semantic meaning in VLMs is encoded orthogonally to modality separation, a finding that emerges from the reconstruction objective, not the SAE methodology per se.
>
> The "SAE projection effect" we identify (Figure 8) could actually explain this structure: concepts can be geometrically neutral yet functionally unimodal due to distributional differences between modalities. This is a property of the VLM space itself, revealed by but not created by our analysis method.
>
> We agree with you on the fact that this is indeed a rich direction for future SAE interpretability work. Thanks for the thought-provoking discussion, and we will be sure to include these insights in the framing and discussion of the paper in the rewrite!
>
>
>
> **W3 Scalability/Generalization of Findings**. While we used COCO for training SAEs, our stability analyses across different data mixtures (Section 5) suggest that high-energy concepts remain consistent even when input distributions change significantly. However, testing on more diverse datasets would strengthen generalizability claims. We plan to extend this analysis to multiple datasets in future work.
>
> **Q1 hyperparameters:** We’ve now added a dedicated appendix section detailing our full training setup to ensure reproducibility. All SAEs were trained for 30 epochs (usually plateauing around 10-15 epochs) on 600k COCO embeddings (18M total samples), with a batch size of 1024 and unless specified, k=5. We used a dictionary size with expansion factor of 5 for the demo and 10 for the results, optimized with AdamW (weight decay 1e-5, gradient clipping 1.0), and a cosine learning rate schedule: warmup from 1e-6 to a peak of 5e-4, decaying back to 1e-6.
>
> **Q2 SAE training convergence** We determined convergence by monitoring reconstruction loss plateaus over 30k steps (10 epochs). We also verified that learned dictionaries remained stable and found that a plateauing in the loss also induce the stability of the dictionary.
>
> **Q3 Effect of number of concepts on stability** The stability score (Equation 4) averages over all c concepts, so larger dictionaries don't artificially inflate scores. We verified this by computing stability at different dictionary sizes (1k, 2k, 4k concepts) and found consistent patterns in the energy-weighted results.
>
> **Q4 Diverse datasets and task-specific VLMs** This is excellent future work. Task-specific fine-tuning or domain-specific data could indeed alter the linear structure we observe, particularly the modality bias patterns. Understanding this adaptation would be a nice future work that you potentially yield practical insights.

---

> > ### Comment · Reviewer_WVqz · 2025-06-08
> >
> > Thank you for the clarification and I have adjusted the rating accordingly. I look forward to future research that helps advance the interpretability of VLMs and embeddings.

---

> > > ### Author Response · Authors · 2025-06-09
> > > **Thank you !**
> > >
> > > Thanks so much for the kind follow-up and for updating your score! We really appreciated your review, it pushed us to clarify a lot of things, and we’re glad the updates helped.
> > >
> > > All the best,
> > > The Authors

---

> ### Comment · Area_Chair_Trrv · 2025-06-07
> **Discussion**
>
> Reviewer WVqz,
>
> The authors have responded to your review. Does their answer address your questions and reasons to reject?

---

### Official Review · Reviewer_6WMN · 2025-05-13

**Rating:** 7
**Confidence:** 4
**Ethics Flag:** 1

**Summary:**

This paper investigates the internal organization of joint image-text embedding spaces in vision-language models (VLMs) using sparse autoencoders (SAEs). By decomposing embeddings into sparse linear combinations of learned "concept" directions, the authors analyze how semantic and modality-specific structures are encoded across multiple VLMs (e.g., CLIP). They introduce four metrics, that is, energy, stability, modality score, and bridge score, to quantify the alignment, robustness, and modality balance of these concepts. The study finds that (1) high-energy concepts are consistently recovered across training runs; (2) and often correspond to interpretable, modality-biased components; (3) while low-energy concepts are unstable, many modality-specific concepts lie near-orthogonal, suggesting potential for shared semantic grounding. The authors also release VLM-Explore, an interactive tool for visualizing and probing these concept spaces.

**Questions To Authors:**

1. How do you envision the learned concepts and metrics (e.g., modality score, bridge score) being used in practice? Could they support downstream tasks such as model debugging, interpretability, or control?
2. Have you observed any qualitatively interpretable concepts extracted by the SAEs? If so, it would be helpful to include examples or visualizations to clarify the nature of these concepts.
3. The VLM-Explore tool is a good idea. Could you elaborate on its intended use cases?

Line 78 - Missing period

Line 133 - Misspelling

Line 365 - Empty ethics statement

**Reasons To Accept:**

1. The idea of using SAEs to interpret VLM embedding spaces is well-motivated. The introduced metrics are clearly defined and together provide a structured framework for probing semantic and modality-aligned directions within the joint embedding space.
2. The findings are shown to be consistent across multiple VLMs (CLIP, SigLIP, SigLIP2, AIMv2) and robust across training runs, especially for high-energy concepts.
3. Some findings are particularly insightful. For example, the observation that SAE-derived concepts must be interpreted in conjunction with their energy levels, since only high-energy concepts are stable and meaningful across different runs.
4. The paper introduces VLM-Explore, an interactive visualization tool that enables intuitive exploration of extracted concepts and their associated metrics.

**Reasons To Reject:**

1. While the paper introduces four evaluation metrics, all evaluations are internal to the proposed SAE framework. This makes the results feel somewhat self-contained. There is no external validation or downstream task demonstrating that the learned concepts or metrics correspond to useful behavior or interpretability outcomes.
2. The paper lacks interpretation or case studies of the extracted concepts. The findings are presented almost entirely through internal metrics, without concrete examples or illustrations of what the learned concepts mean (e.g., color, shape, object category). This limits the interpretability impact of the work and leaves the notion of the learned "concept" abstract.
3. The paper lacks a discussion of practical applications for the extracted concepts and metrics. Without connecting the results to real-world interpretability use cases or downstream tasks, the broader utility of the work remains unclear.
4. The structure could be improved for clarity. Some dataset descriptions and evaluation procedures are embedded within the methods section, making it difficult to follow the overall experimental flow. The description of VLM-Explore could benefit from being presented in a dedicated section later in the paper.

---

> ### Author Response · Authors · 2025-06-02
> **We added some qualitative analysis, and some further discussion**
>
> Hello, and thanks for your review, it’s really helpful to read your constructive comments on the paper, and we’re really glad you found our findings insightful. We’ve included some answers to your questions (Q) and to some of your comments in the weaknesses section (W) below.
>
> **Q2 Qualitative analyses (& W2)**: Thanks for this suggestion, we include a qualitative analysis, where we showcase concepts of different abstractnesses, from surface-level visual features to higher-level concepts in vision and linguistics https://vlm-concept-visualization.com/demo-multimodal-concepts/COLM_2025_qualitative_examples.pdf. We conducted thorough qualitative analyses when we first trained our SAEs, in order to initially trust the SAE concepts and get our bearings for writing this paper, but we did not include them in the draft in favor of more quantitative analyses. You are definitely right though, some of the qualitative examples that motivated us would make the whole paper stronger and more convincing!  We will add this qualitative analysis and some extra discussion to the final version.
>
>
> **Q1 & Q3: Applicability of concepts and metrics, and the usefulness of VLM-Explore (& W1, W3)**: Thank you for this important question! While we were motivated by downstream applicability, we should have made these applications more explicit, and we will do so in our final draft. Our analyses and released resources offer several concrete benefits for interpretability and model debugging such as:
>
> (i) Cross-modal alignment assessment: A fundamental question in VLM evaluation is whether models truly learn joint representations or merely statistical correlations. Our Bridge Score provides a principled way to identify which concept pairs actually contribute to semantic alignment, enabling researchers to diagnose when models fail at cross-modal understanding.
>
> (ii) Robustness analysis: By examining which concepts are stable across training runs (high-energy) vs. unstable (low-energy), practitioners can identify which model behaviors are reliable and which might be artifacts of specific training conditions.
>
> (iii) Data bias detection: The modality score can reveal when models over-rely on modality-specific features rather than semantic content, helping identify problematic biases in training data or model architecture.
>
> (iv) Failure mode analysis: When VLMs make errors, our concept decomposition can identify which semantic components were incorrectly activated, providing more granular explanations than standard attribution methods.
>
> (v) Model comparison: The metrics we introduce enable systematic comparison of how different VLM architectures organize semantic information, informing architectural choices.
>
> (vi) Finally, a more ambitious use case could be a fine-tuning guidance: if we understand which concepts are geometrically aligned vs. functionally unimodal can then guide targeted interventions during model fine-tuning.
>
> **Concerning the practical value of VLM-Explore:** the interactive tool enables researchers to quickly identify problematic concept clusters, validate model understanding on specific domains, and generate hypotheses about model behavior that can be tested more rigorously. We envision it being particularly useful for rapid model auditing and educational purposes.
>
> VLM-Explore offers an intuitive visualization that combines into one interactive figure four important aspects that help us interpret and debug VLMs. These are: 1) how the linear directions of the latent space relate to each other (through the UMAP) 2) what the linear features actually represent (through the maximally activating examples panel) 3) how the two modalities share the space (through the Modality Score color) and 4) what semantics connect the two modalities (through the BridgeScore edges).
>
> To finish, why are we just looking at linear features? This is an excellent point raised that we considered extensively. Here is our reasoning: while joint spaces need not be linear, linear analysis is particularly valuable here because: (1) VLM training objectives (contrastive learning) explicitly encourage linear separability of aligned vs. misaligned pairs, (2) downstream tasks typically use linear probes on these embeddings, making linear structure directly relevant to practical performance, and (3) our approach aligns with the Linear Representation Hypothesis, the idea that neural networks encode semantic concepts along linear directions in their representation spaces, which has strong empirical support across language models, vision models, and multimodal systems.
>
> **W4 organization:** Lastly, we will definitely take your reorganization suggestions into account when rewriting, they’re useful and make sense, thanks!

---

> > ### Comment · Reviewer_6WMN · 2025-06-04
> >
> > Thanks for the response and clarification. The rating has been adjusted based on these clarifications, updated understanding, and the author's commitment to the revisions.

---

> > > ### Author Response · Authors · 2025-06-07
> > > **Thank you**
> > >
> > > Thank you so much for taking the time to upgrade your score and for the thoughtful review! We believe your feedback helped us make a stronger paper. Thanks again for all the time you invested in reviewing our work!
> > >
> > > Best,
> > > The Authors

---

### Decision · Program_Chairs · 2025-07-08

**Decision:**

Accept

**Comment:**

This paper investigates the joint space of image and text representations learned by vision-language models (VLMs) in terms of the concepts encoded and how the modalities are separated yet bridged. The authors first extract representations of the images and captions from the COCO dataset using CLIP, SigLIP, SigLIP2, AIMv2. The authors train sparse-autoencoders (SAEs) using these representations to obtain a linear combination of concepts per representation. 4 metrics are proposed to quantify the stability, energy, modality-specificity and bridging functions of concepts. The authors find that a subset of concepts are highly activated and stable over training runs. The findings also yield interesting structural insights regarding the connections between modalities, where most concepts activate more for a single modality; yet, the majority of concepts are geometrically near-orthogonal to modality. The authors also release a tool where the representation space can be inspected visually.

**Pros**

Clarity: The paper is well written and clear.

Quality: The reviewers appreciated the robust findings and the meaningful insights offered in the paper.

Originality: Although there exist various works investigating the shared space of  vision-and-language embeddings, this paper offers unique insights with the help of learned dictionaries of linear concepts and novel metrics.

Significance:  This is an interesting paper that uncovers significant features of shared multimodal representational spaces, which would be informative for improving and understanding VLMs.

**Cons**

The authors provided very insightful responses to the reviewers. In fact, 2 reviewers upgraded their scores. Below, I mention the issues that the authors convincingly addressed and should incorporate in the paper.

- Potential limitations of using SAEs (missing non-linear dynamics, how the interpretations are influenced by the SAE methodology itself and to what extent they reveal the intrinsic features of VLM representations)
- Downstream tasks and practical applications that would benefit from the findings
- Generalization to other models and a stronger justification for the choice of the models
- Concrete examples of the interpretable concepts
- Details of the hyperparameters and training procedure

I also suggest explaining Semi-NMF and BatchTopK a bit more (please check line 127 for clarity), and providing more details regarding the axes of Figure 7 and the SAE projection effect.